# Grape Pomace Fibres as a Sustainable Fining Agent to Ensure Red Wine Safety: A First Approach in a Continuous System

**DOI:** 10.3390/foods14091565

**Published:** 2025-04-29

**Authors:** Lucía Osete-Alcaraz, Encarna Gómez-Plaza, José Oliva-Ortiz, Miguel Ángel Cámara, Bodil Jørgensen, Ricardo Jurado-Fuentes, Ana Belén Bautista-Ortín

**Affiliations:** 1Department of Food Science and Technology, Faculty of Veterinary Sciences, University of Murcia, 30100 Murcia, Spain; lucia.osetea@um.es (L.O.-A.); anabel@um.es (A.B.B.-O.); 2Department of Agricultural Chemistry, Faculty of Chemistry, University of Murcia, 30100 Murcia, Spain; josoliva@um.es (J.O.-O.); mcamara@um.es (M.Á.C.); 3Department of Plant and Environmental Sciences, University of Copenhagen, DK-1165 Copenhagen, Denmark; boj@plen.ku.dk; 4Agrovin S.A., Avenida de los Vinos s/n, 13600 Alcázar de San Juan, Ciudad Real, Spain; rjurado@agrovin.com

**Keywords:** OTA, histamine, pesticides, red wine, fining, continuous filtration, plant fibres

## Abstract

Grape pomace is the largest by-product in the oenological industry, and in recent years, there have been multiple attempts to turn it into a high-value product, such as a fining agent. However, most of these attempts have usually been conducted with low volumes of wine, and/or in static conditions, using a long contact time between the fibre and wine. To speed up the fining process, this study evaluated the effectiveness of three pomace fibres and a commercial fibre in improving the safety of a young red wine, previously contaminated with ochratoxin A, histamine, and various pesticides, using a continuous filtration system. All the pomace fibres were capable of reducing the OTA concentration by around 50%, and one of the tested fibres exhibited a strong ability to decrease most of the pesticides present in the wine, with the results being even better than when this fibre was used in static conditions. All the tested fibres similarly reduced the tannin concentration of the wines, without having a major impact in the colour index. These results prove that pomace grape fibres are an effective fining agent suitable for use in a continuous filtration system, allowing for a reduction in the fining time from days to hours.

## 1. Introduction

Ensuring the sustainability of the winemaking process is a continuous challenge to which new answers are daily emerging. One of the biggest by-products in the enology industry is grape pomace. Each year, grape crops yield 63 million tonnes worldwide, and 75% of these crops are dedicated to wine production [1]. Approximately 20% of the grapes’ weight constitutes grape pomace, the main winemaking by-product [2]. In Spain alone, 5 million tonnes of fresh grapes were produced in 2023, making it the third largest wine-producing country in the world (OIV) [3].

Grape pomaces have traditionally been re-used as low value-added products, such as animal feed, compost, or bioenergy [4]. In recent years, numerous studies have been conducted to explore the use of this by-product as a new fining agent [5,6,7,8], aiming to turn it into a valuable product that can be reused in the same industry. Furthermore, there has been increasing interest in developing effective fining agents of vegetal origin to replace the traditional fining agents, such as casein, gelatin, albumin, and isinglass, which have an allergenic potential and are not suitable for a vegan diet.

Wine fining consists of adding a substance or a mixture to stabilize or modify the organoleptic characteristics of the wine, but it could also be used to enhance the safety of wine without modifying its colour and flavour. In terms of possible wine contaminants, some may be produced during the winemaking process and others could be carried over from the vineyard itself. In this study, we wanted to focus our attention to three types of contaminants of different origins and chemical nature: pesticides, ochratoxin A (OTA), and histamine.

The presence of pesticides in wines has been detected and reported in many countries in the Mediterranean area [9,10,11,12]. Fungicides are generally the most used pesticides due the common presence of moulds in the vineyard, such as *Botrytis cinerea* (gray mould), *Uncinula necator* (powdery mildew), and *Plasmopara viticola* (downy mildew), which are the main infections that affect grapes [13]. The effect of pesticides on human health is difficult to determine because it depends on several factors, and humans are exposed to them through various routes, like residues in food and drinking water [14]. However, because many of these pesticides have the potential to bioaccumulate [15], the continuous exposure to pesticides has been related to multiple illnesses and disorders [16]. To date, maximum pesticide limits have not been established for wine; these limits are only set for grapes within the European Union (European Commission, 2005).

Ochratoxin A is a mycotoxin whose presence in wine is linked to the fungal contamination of grapes, mainly by *Aspergillus* and *Penicillium* [17]. OTA was classified as a potential human carcinogen (group 2B) by the International Agency for Research on Cancer (IARC), with nephrotoxic, hepatotoxic, teratogenic, and carcinogenic effects [18,19], and it has been correlated with tumours in the human urinary tract [20,21,22]. To date, OTA has been detected in wheat, corn, cocoa, nuts, and malt, and therefore, it has also been found in final products, such as beer, wine, chocolate, and bread [19,23,24]. However, the processing techniques may vary the amount of OTA in the final product. In the case of wine, it has been proven that long macerations increase OTA’s final concentration in this product [25]. Due to health-related concerns, the maximum tolerable level in wines was established by the European Commission (regulation 1881/2006) at 2 µg/L.

The main biogenic amines (BAs) found in wine are histamine, tyramine, putrescine, cadaverine, and phenylethylamine [26,27], and their main origins have been related to wine malolactic fermentation and wine storage [28,29]. While low levels of these amines are normal in the body, an excessive intake of them could lead to health problems, such as itching, headache, hypertension, and tachycardia [30]. Histamine is the BA found in the highest concentrations in wine. In the last decade, histamine intolerance has gained recognition, with most patients manifesting gastrointestinal problems, abdominal distention, and pain, followed by dizziness, headaches, and palpitations [31]. However, it is important to note that high histamine concentrations will not only affect sensitive populations but all consumers. Some European countries have already established maximum limits of histamine in wine: 2 mg/L in Germany, 5 mg/L in Finland, 10 mg/L in Australia and Switzerland, 8 mg/L in France, 3.5 mg/L in Netherlands and 5–6 mg/L in Belgium [32,33].

To this day, most studies on fining with grape pomace fibres have been conducted using small volumes of wine, in static conditions, and with long contact times. With the intention to move closer to a winery approach, accelerate the fining process, and test whether this type of fibre can be used as a high-value product to ensure increased wine safety, in this study, we carried out a continuous-flow experiment. We used an experimental column through which we continuously passed wine spiked with OTA, histamine, and various selected pesticides. Fibres from three different grape varieties were tested as fining agents, and the results were compared with a commercial plant fibre.

## 2. Material and Methods

### 2.1. Fining Trial

The fining experiment was conducted using four plant fibres, at 2 g/L, the highest concentration allowed by the OIV (OIV-OENO 684A-2022): two white grape pomaces of different varieties (WGP1 and WGP2), a red grape pomace (RGP), and a commercial fibre of unknown origin (CoF) (Flowpure, Laffort, 33072, Bordeaux Cedex, France). However, previous studies have shown that the latter fibre presents a different polysaccharide composition compared to those from grape [8]. WGP1 and WGP2 were both provided by Agrovin S.A. (Alcazar de San Juan, Ciudad Real, Spain), and RGP was obtained by purifying a pomace supplied by Explotaciones Hermanos Delgado SL (Socuellamos, Ciudad Real, Spain). All the grape pomace fibres were washed with a hydroalcoholic solution (70%) for 48 h on a shaker at 250 rpm. Afterwards, they were dried at room temperature and crushed in an electric grinder with blades. Then, all the plant fibres were passed through a 50 μm aperture sieve. The three grape fibres came from different regions of Spain. It was considered appropriate to select different fibres to test if the different vinification methods the pomace went through (no fermentation in the case of white grape pomace and maceration–fermentation in the case of red grape pomace) could affect their future behaviour.

A young *Monastrell* wine provided by Bodegas Salzillo (Jumilla, Spain), with a pH of 3.92, alcohol content of 13.5% and a total acidity of 4.5 mg/L, was spiked with OTA (4 µg/L), histamine (10 mg/L), and 7 pesticides, 6 fungicides, and 1 insecticide (Table 1). The wine was passed through a filter in a column composed of diatomaceous earth (Radifil RP, Agrovin S.A, Alcazar de San Juan, Ciudad Real, Spain) (9 g) and the tested fibre (Figure 1). The wine was pushed through the column by a pump machine at a flow rate of 25 mL/min. For each treatment, 1 litre was used, and four fractions of 250 mL were collected during the filtration process (F1, 2, 3, and 4). After the fining treatment, the wine was filtered (0.45 µm nylon) before analyses.

### 2.2. Chromatic Parameter and Phenolic Compound Determination

The total anthocyanins (TAs) and polymeric anthocyanins (PAs) were determined following the described methodology [34]. The total tannins were measured by the precipitation method with methylcellulose (TT) [35], and the colour intensity (CI) was obtained by summing the absorbances at 620 nm, 520 nm, and 420 nm [36].

### 2.3. OTA Determination

To detect ochratoxin A, the Biosystems Ochratoxin A/Competitive ELISA Kit (ref: 14108) was employed. This kit allows for quantitatively detecting OTA through an enzyme-linked competitive immunoassay, comparing the absorbance at 450 nm of the samples with a calibration curve obtained using the standards included in the kit. The calibration range was from 0 to 1 µg/L, so a dilution of 1:4 was needed.

### 2.4. Histamine Determination

The determination and quantification of histamine were determined through an enzyme-linked competitive immunoassay using LDN HistaSure ELISA Fast Track (ref: FC E-3600). This kit allows for detecting and quantifying the histamine concentration using their standards to establish a calibration curve. The samples were measured at 450 nm. The calibration range was from 2.5 to 250 mg/L.

### 2.5. Pesticide Determination

The pesticides were analysed following the QuEChERS protocol [37] on a TQ QUANTIS^®^ liquid-mass HPLC (LC-MS). First, 10 g of the sample was homogenized with 10 mL of acetonitrile (ACN). After the addition of partitioning salts (anhydrous magnesium sulphate, sodium chloride, disodium citrate sesquihydrate, and trisodium citrate dihydrate) and centrifugation of the extract, the supernatant was transferred to a vial and acidified with 5% formic acid in ACN, preparing it for chromatographic injection. The chromatographic analysis was carried out using liquid chromatography coupled with a triple quadrupole mass spectrometer (LC-MS/MS) equipped with an electrospray ionization (ESI) source operating in positive mode. The used column was the Poroshell 120 EC-C18 column (3.0 × 100 mm, 2.7 μm), maintained at 40 °C, with a flow rate of 0.6 mL/min and a sample injection volume of 5 μL. There were two mobile phases, solvent A (0.1% *v*/*v* formic acid in ACN) and solvent B (0.1% formic acid and 2 mM ammonium formate in water). The gradient started with 20% solvent A at the time of injection, which increased linearly to 100% by 10 min, and then returned to the initial conditions. At the beginning and end of the analysis, duplicate samples were included as quality controls to assess sample stability.

### 2.6. Comprehensive Microarray Polymer Profiling (CoMPP) Analysis of the Fibres

For analysis of the composition of the different fibres, CoMPP was used. This technique enables the detection of glycan epitopes in plant tissues using monoclonal antibodies (mAbs) and carbohydrate-binding modules (CBMs) [38]. Polysaccharide extractions were obtained by weighing 10 mg of alcohol-insoluble residue (AIR) and treating it with 600 μL of 50 mM trans-1,2-diaminocyclohexane-N,N,N′,N′-tetraacetic acid (CDTA), followed by 4 M NaOH containing 0.1% (*v*/*v*) NaBH_4_. Each extraction was shaken at 6 s^−1^ for 2 h, followed by 10 min centrifugation at 4000 rpm and the collection of the supernatant. CDTA mainly allows for the extraction of pectins, especially those highly exposed, due to the removal of Ca^2+^ ions, and NaOH targets hemicelluloses, unbranched RG-I, galactans, xyloglucans, and strongly associated pectins, like highly esterified HG and RG-I [39]. Each extraction was carried out in triplicate.

A piezoelectric array printer (Marathon, Arrayjet, UK) was used to create a profile array of the samples, which was later probed with primary antibodies specific to the plant cell wall components, followed by secondary antibodies conjugated to alkaline phosphatase. The microarrays were developed using BCIP/NBT (5-bromo-4-chloro-3′-indolyl phosphate/nitro-blue tetrazolium chloride). The measuring and quantification of the intensity of the arrays was carried out using ImaGene 6.0 microarray analysis software. The highest signal was set to 100 and any signal below 5 was recorded as 0. Finally, the processed data were visualized as a heatmap using Excel software (version 2503).

### 2.7. Statistical Analysis

The data were statistically processed using a one-way analysis of variance (ANOVA) followed by an LSD (least significant difference) test (*p* < 0.05) in STATGRAPHICS Centurion XVI.3 (Statpoint Technologies, INC., The Plains, VA, USA) software.

## 3. Results and Discussion

### 3.1. OTA Reduction

The results obtained by the ELISA analysis show that all the fibres were capable of significantly reducing the OTA content in the wine, by at least 50% (Table 2). The commercial fibre (CoF) exhibited the greatest ability to reduce OTA, by around 90%, while the pomace fibres showed a more limited ability, between 50 and 66%. RGP was proven to be statistically more effective at reducing OTA than WGP1 and WGP2. These results contrast with those obtained in a previous experiment [40], in which CoF showed a similar performance in reducing OTA, but WGP1 (called SBF in that work) had a lower capacity to reduce this compound in red wine, by 30%. It is important to note that this experiment was carried out under different circumstances, that is, static conditions for 48 h. Therefore, it seems that by using pomace fibres in a continuous filtration system, we can not only shorten the fining time but also achieve a greater reduction in OTA. However, it is important to note that some of the OTA reduction could be attributed to the diatomaceous earth, as in a comparable test in which the filter was composed of only the amount of diatomaceous earth used in the fibre filters (9 g), 11% of OTA was removed. These results are also comparable to other results published [41], that reported an OTA reduction of 54–57% using fibres from grape origin. Nevertheless, in this work, higher concentrations of fibres were used (13 mg/mL) under static conditions, and the contact time lasted 21 days.

Multiple studies have been conducted using the most common fining agents, such as gelatin, egg albumin, isinglass, and bentonite [41,42,43,44]. Most of them reported lower levels of OTA reduction (10–34%) than those reported in this study, and often with a high impact on the wine chromatic characteristics.

These results indicate that grape pomace fibres are an interesting OTA removal agent, capable of eliminating high quantities of OTA in wines, possibly due to their complex polysaccharide composition that provides numerous binding sites, allowing both hydrogen bonds and ionic interactions with OTA, which is a compound negatively charged at wine pH (3.2–3.6).

### 3.2. Histamine Reduction

The histamine reduction achieved by all the tested fibres was notorious (Table 3). The one that achieved the greatest reduction was WGP2, with a 27% reduction, although its effect was not significantly different to that of CoF and WGP1, which led to histamine reductions of 20% and 16%, respectively. RGP was the least effective in reducing histamine, by just 12%, although this reduction was still statistically different with respect to the control wine. In a previously conducted static test [40], WGP1 achieved a higher histamine reduction, of 36%, whereas CoF led to a reduction of 12% of histamine content. In comparison, under the conditions used in this study, no significant differences in their reduction capacity were observed. Analysing the histamine reduction in F1, 2, 3, and 4, the fibres did not all behave in the same way. The reductions achieved by WGP1 and RGP were constant from F1 to F4, while WGP2 and CoF were much more effective in reducing histamine in F1 (the first 250 mL of wine recovered after passing through the column), than the rest of the fractions. WGP2 reduced its effectiveness in F2, F3, and F4, but the histamine reduction level of these fractions was still between 21% and 27%, while CoF exhibited histamine levels in F3 and F4 similar to the control, indicating quick saturation of this fibre.

We could only find one other study where fibres derived from grape pomace were used to reduce histamine in red wine [41], and they reported histamine reduction levels of 10%, even when they used higher concentrations of fibre over a longer contact time. Histamine is positively charged at wine pH, which could allow it to bind with different fining agents by ionic exchange or by hydrogen bonding type interactions [45].

### 3.3. Pesticide Reduction

The pesticide reduction capacity of each fibre was measured using QuEChERS-LC-TSQ. The pesticides used in this study were six fungicides (iprovalicarb, fenhexamide, boscalid, tetraconazole, mepanipyrim, and metrafenone) and one insecticide (imidacloprid).

In general, the statistical analysis showed that for fenhexamide, imidacloprid, and iprovalicarb, all fibres produced similar reductions in their content in wine, and they were less absorbed in the fibres than boscalid, mepanipyrim, metrafenone, and tetraconazole (Table 4). For these latter pesticides, WGP1 and CoF showed similar behaviour in their retention capacity, with these two fibres being more effective than RGP and WGP2.

In the case of fenhexamide, and for WGP1 and CoF, a larger reduction of this fungicide was found in F1, with a reduction of over 60% for CoF and over 70% for WGP1, while for RGP and WGP2, the reduction was around 25% in the same fraction. This effect was no longer observed in the following fractions for WGP1 and CoF, where the reduction percentages dropped considerably. This is the reason no statistically significant differences were observed in the overall effect among all fibres for this fungicide. For iprovalicarb, a similar result was observed, although with a lower reduction capacity for all fibres, with CoF and WGP1 presenting a retention of 30% of this pesticide in F1, while for WGP2 and RGP, this decrease was way lower and with high variability. The imidacloprid behaviour was slightly different; CoF exhibited an interesting ability to reduce this insecticide, by around 55% in F1, while WGP2 and WGP1 reduced it by around 25–30%, also in F1. These fibres did not exhibit the same behaviour in the remaining fractions, showing lower retention values of this pesticide with the passage of more wine volume through the filter.

For mepanipyrim and tetraconazole, all the grape pomace fibres exhibited similar behaviour, with a progressive decrease in the capacity of pesticide reduction as the millilitres of wine passing through the column increased. However, for CoF, this decrease only was seen in the last wine fraction. WGP2 and RGP reduced their contents by 50% and 40% in F1, respectively, whereas WGP1 lowered the levels of both pesticides by 90% in the first fraction, resulting in a higher overall adsorption capacity of WGP1 for these two pesticides than WGP2 and RGP.

For boscalid, its retention in F1 ranged between 26% and 83%, showing a similar retention pattern compared to the previous pesticides, although with a more important decrease in the reduction percentage for all fibres as the volume of wine passing through the column increased. The behaviour of CoF was slightly different; its retention capacity only started to decrease at F4. This indicates that this fibre has a higher saturation threshold. For metrafenone, all fibres showed the same adsorption capacity throughout the entire process, with WGP1 being the most effective in its retention, achieving a decrease of 94%. It is important to consider that the effect of diatomaceous earth in a comparable test, in which the filter was composed of only the amount of diatomaceous earth used in the fibre filters (9 g), was minimal for all pesticides except metrafenone, where diatomaceous earth adsorbed 26% of this pesticide, probably due to its high Koc.

To better understand the reasons behind the differences in pesticide adsorption, it is crucial to examine the n-octanol/water partition coefficient (log Kow), the soil adsorption coefficient (Koc), and the water solubility of the pesticides involved (Table 1). The removal of pesticides is strongly influenced by the log Kow, which is closely linked to the Koc, as can be seen in Figure 2. A higher log Kow indicates greater apolarity, meaning a stronger tendency to bind to solid substances and lower solubility in water [46]. Boscalid, fenhexamide, iprovalicarb, mepanipyrim, and tetraconazole exhibited moderate log Kow values (ranging from 3 to 3.5), while imidacloprid had a very low log Kow (<1) and metrafenone a high log Kow (>3.5). It is important to note that even though iprovalicarb had a moderate log Kow (3.19) its Koc was low in comparison with the rest of the pesticides with a similar log Kow (106), and even lower than that of imidacloprid Koc (225). With this understanding, we can explain why iprovalicarb and imidacloprid were the two pesticides less adsorbed by the fibres. However, WGP1 and CoF showed a certain capacity to retain them, especially for the first 250 mL of wine passed through the column (F1), so they can be considered a useful tool to reduce these pesticides. For boscalid, mepanipyrim, tetraconazole, and metrafenone, the reduction capacity seemed to be strongly correlated to their log Kow, which explains why metrafenone showed the greatest affinity for all fibres.

In a previous work already mentioned above [40], WGP1 and CoF already proved their efficacy in removing these pesticides under static conditions, although we observed some differences. WGP1 improved its performance working under continuous conditions, such as those used in this trial, increasing its ability to reduce tetraconazole (from 42% to 68%), mepanipyrim (from 50% to 74%), and metrafenone (from 57% to 94%), while its ability to reduce fenhexamide, imidacloprid, and boscalid remained very similar. Only for iprovalicarb did its performance decrease (from 42% to 22%). This could be due to this pesticide needing more contact time to bind to the fibres. Regarding CoF, it was observed that its performance only improved for mepanipyrim (from 56% to 68%) and metrafenone (from 69% to 77%), while for the rest of the pesticides, its effectivity in reducing their concentrations in wines remained similar, except for iprovalicarb (from 38% to 10%) and fenhexamide (from 43% to 29%), where it decreased.

CoF was used in Austrian red wines as a fining agent in a previous work [47], working in static conditions, with a contact time of 18 h and a dose of 2 g/L. In their study, CoF did not significantly reduce mepanipyrim and fenhexamide, while in our study, we found reductions of 69% and 29%, respectively. Also, they reported a 50% boscalid reduction capacity, while we obtained reduction levels of 66% in our study.

### 3.4. Effects on the Chromatic Parameters and Phenolic Compounds of Wine

Although one of the objectives of fining is often the reduction of wine tannins, mainly because a high concentration of these compounds is related to high wine astringency and bitterness, most of the time, the main concern during the fining step in red winemaking is to achieve effective wine fining without modifying its chromatic parameters to a considerable extent. To check the effect of this fining system on wine colour, we measured the total polyphenol index (TP), colour index (CI), total anthocyanins (TAs), polymeric anthocyanins (PAs), and total tannins (TTs).

The results show that CoF and WGP1 were the only treatments that significantly reduced the CI (Table 5). As can be seen in the data of F1, 2, 3, and 4, F1 was always the most affected in terms of colour loss. This is especially noticeable for CoF, with a CI reduction of 13%. However, F2, F3, and F4 did not produce changes in the values of this parameter with respect to the control. The values of TA and PA exhibited a similar pattern as the CI; the main decrease was also observed in F1, while the other fraction values were similar to the control. Once again, CoF exhibited the highest reductions in these parameters, reducing their content by 16% in F1. Overall, the affinity of the grape pomace fibres for anthocyanins, and therefore, their effect on colour reduction, appeared to be weak, while in the case of CoF, it seems that the fibre was quickly saturated as the wine flowed through it. In other studies where grape pomace fibres were used [6,7], a bigger impact on the chromatic characteristics was observed, with a reduction of around 10–12% in TA, probably due to the use of higher fibre doses (5 and 6 mg/mL).

Tannins are important wine compounds, and usually, they are the compounds effected the most during the fining process with grape pomace fibres. This can be either favourable or detriment to wine quality. For example, with an excess of tannins, the use of fibres could be a tool to reduce the astringency of these compounds.

All the tested fibres reduced the total tannin content, within a range between 22% and 26%, with CoF again having the highest impact on their concentration. The reduction of tannins was consistent, with small variations in the four fractions, indicating that the fibres did not saturate as quickly as they did with anthocyanins. Only when CoF was used, a decrease in the adsorption capacity between F1 and F2, 3, and 4 was observed, although the reduction of tannins in F1 was 36%. All these results match with what was observed in the TP levels, where the greatest reductions occurred with CoF, with this fibre having the most significant effects on the chromatic parameters of the wine.

The ability of WGP1 and CoF to interact and bind the phenolic compounds of red wine was also tested previously [8] in static conditions for 48h, at the same concentration. Under these circumstances, the reduction of tannins was very similar for CoF and slightly lower for WGP1, while both fibres had more impact on the anthocyanin content and consequently on colour. Therefore, it seems that a shorter contact time did not significantly affect the binding of tannins, while it could prevent anthocyanin binding and their consequent loss. Some studies proposed that anthocyanins react and bind with cell walls in two steps, initially via ionic and hydrophobic interactions, in a step that appears to be quick, followed by an apparent slower step consisting of further anthocyanin pile-up in the attached layer [48]. This could explain why the reduction of anthocyanins was lower in this study than in those with longer contact times.

A similar reduction of tannins of 21%, was reported, through methylcellulose precipitation using *Monastrell* grape pomace fibre, with a 6 mg/mL concentration for 5 days of contact [7].

Most previous studies on grape pomace fibres have used longer contact times, higher fibre concentrations, or both. In these conditions, the reduction of tannins achieved was either similar or lower, and there was a higher impact on the wine chromatic characteristics. These results indicate that the continuous filtration system could be the most suitable system to use these fibres with the aim of reducing tannins, as it would allow for reducing the used dose (and in this way, match the dose approved by the OIV), and the contact time.

It is also remarkable that all used grape fibres reduced tannins in a similar way, which contrasts with previously reported results [6,7], where tannin reduction was more influenced by the grape variety used to obtain the fibre.

### 3.5. Effects of the Fibres Polysaccharide Composition on the Reduction of Undesirable Compounds

The different fibres presented significant differences in their ability to reduce histamine and OTA. However, these differences were small compared to the differences found in reducing pesticides, specifically, the higher capacity of WGP1 to reduce the studied pesticides in comparison to RGP and WGP2. Regarding the chromatic parameters, the three fibres from grape pomace behaved similarly. One of the main questions that arose when analysing the data is what could explain the differences between the fibres’ behaviours. To try to answer this question, we looked at the fibre polysaccharide composition, which was studied by the CoMPP technique (Figure 3).

The CoMPP results for CoF confirm the information about its composition in its data sheet, with pectins being absent in both fractions. CoF is mainly composed of xyloglucan and xylan (with signals of LM15, LM25, LM10, and LM23 antibodies in the NaOH fraction).

The compositions of WGP1, WGP2, and RGP are very similar in the NaOH fraction, with similar levels of partially methylesterification homogalacturonan I, rhamnogalacturonan I, and mannan (with signals of LM19, INRA-RU1, and LM21, respectively). In the CDTA fraction, the levels of the highest exposed pectins were also very similar among the three fibres, with WGP2 having slightly lower levels of homogalacturonan with low methylesterification (signal of JIM5) than WGP1 and RGP. The main difference that can be seen was provided by the INRA-RU1 antibody, which detects rhamnogalacturonan I (RG-I). The WPG1 levels of this polysaccharide are reported to be twice as high as those in WGP2 and RGP in this fraction, indicating that this branched polysaccharide, composed mainly of a backbone of (1,2)-α-D-galacturonic acid-(1,4)-α-L-rhamnose residues [49], is much more exposed in this fibre. These branches could provide a greater number of binding sites for the pesticides, which could be the reason for the observed results.

Regarding CoF, with only the CoMPP results, it is difficult to explain its ability to bind to so many compounds and at high levels. In a previous study [8], not only the polysaccharide composition of this fibre by CoMPP was studied, but also other parameters, like proteins, phenolic compounds, uronic acids, and cellulosic and non-cellulosic glucose. In this study, it was observed that one of the main components of this fibre is cellulose, which was found in much higher concentrations than in WGP1. Also, the levels of lignin of CoF are lower than those of WGP1, which may decrease its cell wall rigidity, increasing its interaction capacity with different compounds, as was reported previously [5]. Therefore, the different polysaccharide compositions observed for the CoF and WGP1 fibres may explain their interaction with the different contaminants.

## 4. Conclusions

In this study, for the first time, we aimed to investigate and better understand the potential of using grape pomace fibres, working in a continuous system, as a fining agent to reduce wine contaminants, such as pesticides, OTA, and histamine, while also taking into consideration the impact they have on wine chromatic parameters and phenolic compounds. Out of all the tested fibres, CoF, followed by RGP, was the most successful in reducing OTA, while CoF, WGP1, and WGP2, especially the latter, reduced the highest concentration of histamine in these working conditions. WGP1 and CoF demonstrated a remarkable ability to reduce most of the used pesticides. Both of these fibres have previously been tested in static conditions, and overall, in this study, they proved to be equally or more effective in lowering contaminant levels, with a similar reduction of tannins, although CoF had a higher impact on wine colour than WGP1. The probable reason the continuous system is, overall, more effective than static conditions, is that the wine is pushed through the full amount of fibre present in the filter, whereas in static conditions, once the fibre precipitates, the interactions with the different wine components become very limited. However, in continuous systems, one important condition that will need to be optimized is the quantity of fibres within the filtration system and the flow rate of the wine through this system, which may influence the retention capacity of certain compounds.

This study shows that grape pomace-derived fibres, a waste product of the oenological industry, seem to be a promising fining agent capable of ensuring red wine safety, without producing significant changes in its quality. It was found that these fibres can be applied not only in static, but also in continuous filtration systems, similar to those used in the wine industry, allowing for a reduction in the fining time from days to hours. Further studies increasing the volume of the system and optimizing the flow rate will be carried out to reach an industrial scale of operation.

## Figures and Tables

**Figure 1 foods-14-01565-f001:**
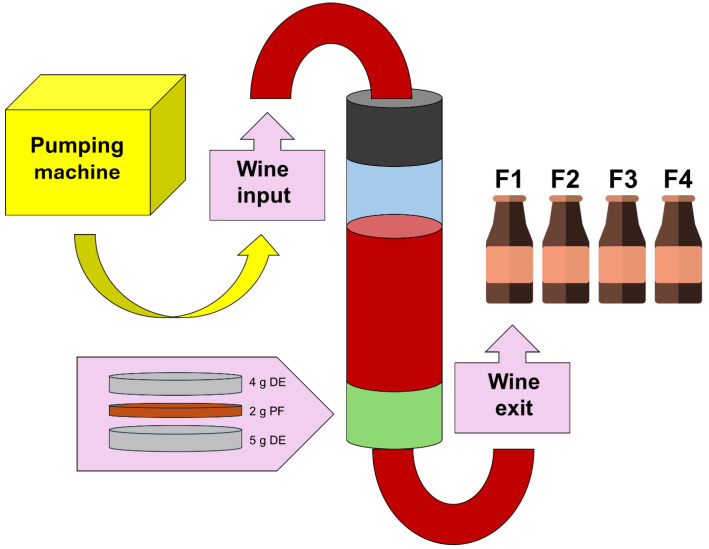
Scheme of the system used to perform fining in continuous conditions. Wine is pumped into a column by a pumping machine. The column has two valves, one at the entry and one at the exit. Before the exit valve, a filter with plant fibres is set up, consisting of three layers: a base layer of 5 g of diatomaceous earth, 2 g of fibre, and 4 g of diatomaceous earth on top. Red symbolizes wine flow; the green valve contains the filter while the black valve only controls the entrance of wine in the system. Abbreviations: DE, diatomaceous earth; PF, plant fibre.

**Figure 2 foods-14-01565-f002:**
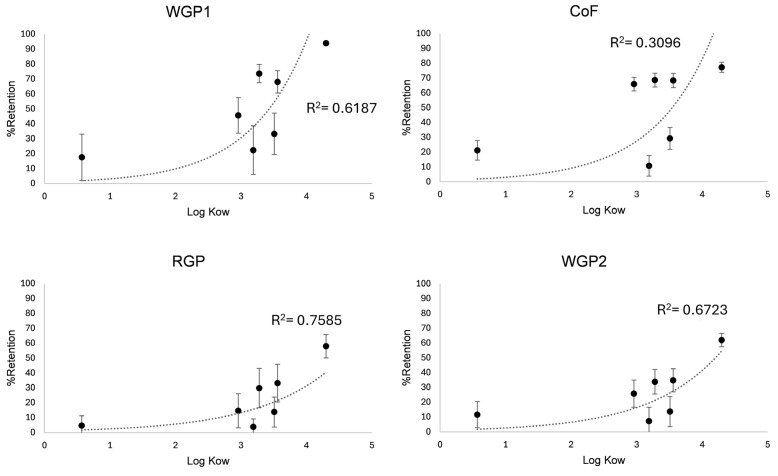
Effects of log Kow on pesticide removal during the fining of a red wine through a continuous filtration system using different plant fibres. Abbreviations: WGP1, white grape pomace fibre 1; WGP2, white grape pomace fibre 2; RGP, red grape pomace fibre; CoF, commercial fibre.

**Figure 3 foods-14-01565-f003:**
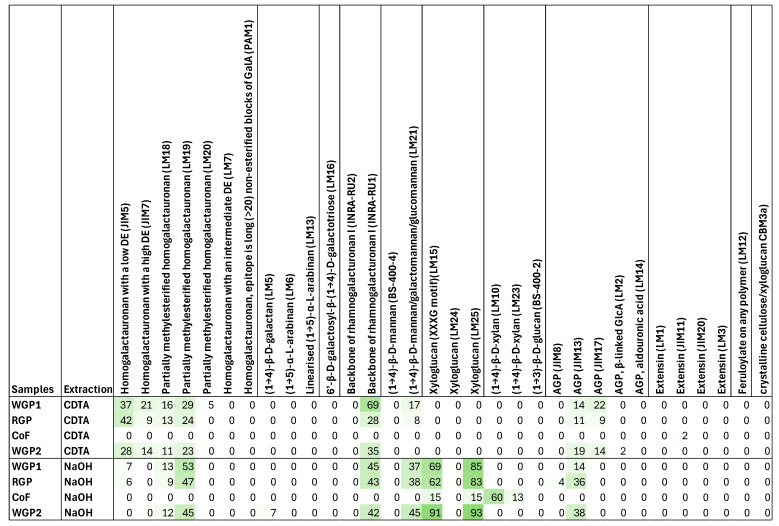
Heatmap of glycan epitope prevalence (0–100) found in the CDTA and NaOH fractions (which mainly target pectins and hemicellulose, respectively) extracted from the cell walls of the plant fibres. Abbreviations: WGP1, white grape pomace fibre 1; WGP2, white grape pomace fibre 2; RGP, red grape pomace fibre; CoF, commercial fibre; DE, methyl esterification degree; GalA, galacturonic acid; AGP, arabinogalactan protein.

**Table 1 foods-14-01565-t001:** Concentrations used of each pesticide, their chemical groups, water solubility, log Kow (pesticide hydrophobicity) and Koc (soil adsorption coefficient).

Pesticide	Chemical Group	Type	Solubility (mg/L)	Log Kow	Koc	Concentration (mg/Kg)
Imidacloprid	Neonicotinoid	Insecticide	610	0.57	225	0.7
Iprovalicarb	Valinamidecarbamate	Fungicide	17.8	3.19	106	2.0
Fenhexamide	Anilide fungicide	Fungicide	20	3.51	446–1226	5.0
Boscalid	Pyridinecarboxamide	Fungicide	4.6	2.96	507–1110	5.0
Tetraconazole	Triazole	Fungicide	156	3.56	531–1922	0.5
Mepanipyrim	Aminopyrimidines	Fungicide	3.1	3.28	875	2.0
Metrafenone	Benzophenone	Fungicide	0.49	4.30	7061	5.0

**Table 2 foods-14-01565-t002:** Fining effect of using different plant fibres in a continuous filtration system on OTA (µg/L) in a red wine. Abbreviations: WGP1, white grape pomace fibre 1; WGP2, white grape pomace fibre 2; RGP, red grape pomace fibre; CoF, commercial fibre. Statistical analysis: ANOVA and LSD test (*p* < 0.05). Different letters indicate statistical differences.

	OTA (µg/L)
**Control**	3.8 ± 0.1 d
WGP1F1	0.0 ± 0.0
WGP1F2	1.9 ± 0.0
WGP1F3	2.7 ± 0.3
WGP1F4	2.3 ± 0.3
**WGP1 mean value**	1.7 ± 0.2 c
RGPF1	0.9 ± 0.3
RGPF2	1.0 ± 0.2
RGPF3	1.5 ± 0.2
RGPF4	1.8 ± 0.1
**RGP mean value**	1.3 ± 0.2 b
CoFF1	0.0 ± 0.0
CoFF2	0.0 ± 0.0
CoFF3	0.6 ± 0.0
CoFF4	1.0 ± 0.1
**CoF mean value**	0.4 ± 0.0 a
WGP2F1	1.5 ± 0.1
WGP2F2	2.0 ± 0.1
WGP2F3	2.0 ± 0.1
WGP2F4	2.0 ± 0.1
**WGP2 mean value**	1.9 ± 0.1 c

**Table 3 foods-14-01565-t003:** Fining effect of using different plant fibres in a continuous filtration system on histamine (mg/L) in a red wine. Abbreviations: WGP1, white grape pomace fibre 1; WGP2, white grape pomace fibre 2; RGP, red grape pomace fibre; CoF, commercial fibre. Statistical analysis: ANOVA and LSD test (*p* < 0.05). Different letters indicate statistical differences.

	Histamine (mg/L)
**Control**	15.0 ± 1.2 c
WGP1F1	13.1 ± 0.9
WGP1F2	12.3 ± 1.1
WGP1F3	12.4 ± 0.1
WGP1F4	12.7 ± 2.0
**WGP1 mean value**	12.6 ± 1.0 ab
RGPF1	14.2 ± 0.4
RGPF2	12.8 ± 2.1
RGPF3	11.5 ± 2.4
RGPF4	14.2 ± 0.4
**RGP mean value**	13.2 ± 1.3 b
CoFF1	9.0 ± 0.7
CoFF2	11.0 ± 0.6
CoFF3	14.3 ± 1.9
CoFF4	13.7 ± 1.4
**CoF mean value**	12.0 ± 1.1 ab
WGP2F1	9.4 ± 1.1
WGP2F2	11.7 ± 0.7
WGP2F3	11.0 ± 1.0
WGP2F4	11.9 ± 0.3
**WGP2 mean value**	11.0 ± 0.8 a

**Table 4 foods-14-01565-t004:** Fining effects of using different plant fibres in a continuous filtration system on pesticides (represented as percentage retention) in a red wine. Abbreviations: WGP1, white grape pomace fibre 1; WGP2, white grape pomace fibre 2; RGP, red grape pomace fibre; CoF, commercial fibre. Statistical analysis: ANOVA and LSD test (*p* < 0.05). Different letters indicate statistical differences.

%Retention	Boscalid	Fenhexamide	Imidacloprid	Iprovalicarb	Mepanipyrim	Metrafenone	Tetraconazole
WGP1F1	83.0 ± 3.9	73.7 ± 4.3	29.1 ± 13.1	36.2 ± 13.3	91.4 ± 1.7	95.9 ± 0.7	90.7 ± 1.7
WGP1F2	38.5 ± 5.7	13.2 ± 12.5	4.3 ± 6.1	7.4 ± 10.5	82.4 ± 2.3	94.4 ± 0.7	78.1 ± 2.7
WGP1F3	28.8 ± 17.3	18.7 ± 12.5	15.1 ± 16.7	18.4 ± 19.5	66.4 ± 6.9	93.6 ± 1.3	56.3 ± 9.6
WGP1F4	31.9 ± 20.6	27.3 ± 20.2	21.5 ± 26.2	27.1 ± 21.8	54.0 ± 13.6	91.6 ± 2.6	46.9 ± 15.8
**WGP1 mean value**	45.5 ± 11.9 bc	33.2 ± 13.9 a	17.5 ± 15.5 a	22.3 ± 16.3 a	73.6 ± 6.1 b	93.9 ± 1.3 c	68.0 ± 7.4 b
RGPF1	26.5 ± 3.7	25.6 ± 3.4	2.0 ± 2.9	1.6 ± 2.2	38.2 ± 5.7	65.0 ± 2.9	43.3 ± 4.7
RGPF2	19.0 ± 24.1	13.9 ± 19.6	10.1 ± 14.3	8.6 ± 12.2	35.4 ± 20.1	54.8 ± 13.8	39.9 ± 18.7
RGPF3	3.7 ± 5.2	5.3 ± 3.1	0.0 ± 0.0	0.0 ± 0.0	29.1 ± 2.6	56.2 ± 2.1	31.1 ± 3.6
RGPF4	9.1 ± 12.8	10.0 ± 14.2	6.4 ± 9.0	4.8 ± 6.8	16.9 ± 23.8	55.7 ± 12.8	18.2 ± 23.7
**RGP mean value**	14.5 ± 11.4 a	13.7 ± 10.1 a	4.6 ± 6.5 a	3.7 ± 5.3 a	29.8 ± 13.2 a	57.9 ± 7.9 a	33.1 ± 12.7 a
CoFF1	76.1 ± 5.0	60.8 ± 5.1	56.2 ± 7.3	31.4 ± 11.7	76.7 ± 3.6	82.5 ± 2.7	78.3 ± 3.5
CoFF2	70.6 ± 0.1	36.7 ± 4.9	18.6 ± 5.0	1.5 ± 2.2	72.7 ± 2.0	80.0 ± 2.2	73.7 ± 1.8
CoFF3	70.3 ± 8.4	19.2 ± 19.7	9.9 ± 14.0	9.8 ± 13.9	71.5 ± 7.7	79.9 ± 4.9	71.2 ± 7.7
CoFF4	46.4 ± 4.8	0.0 ± 0.0	0.0 ± 0.0	0.0 ± 0.0	53.2 ± 5.2	66.3 ± 3.9	49.8 ± 6.1
**CoF mean value**	65.9 ± 4.6 c	29.2 ± 7.4 a	21.2 ± 6.6 a	10.7 ± 6.9 a	68.5 ± 4.6 b	77.2 ± 3.4 b	68.3 ± 4.8 b
WGP2F1	40.9 ± 4.0	27.6 ± 9.0	25.1 ± 5.5.	7.9 ± 8.3	44.2 ± 4.8	66.3 ± 2.6	48.1 ± 4.2
WGP2F2	29.7 ± 19.3	14.1 ± 19.9	13.3 ± 18.8	12.9 ± 18.3	39.8 ± 16.4	62.1 ± 9.0	40.8 ± 16.2
WGP2F3	18.6 ± 5.2	6.3 ± 2.6	3.5 ± 5.0	3.3 ± 4.6	28.3 ± 4.3	60.9 ± 3.1	28.4 ± 4.1
WGP2F4	13.8 ± 8.2	6.4 ± 9.0	4.3 ± 6.1	4.6 ± 6.5	22.8 ± 7.7	58.2 ± 3.5	21.5 ± 6.7
**WGP2 mean value**	25.7 ± 9.2 ab	13.6 ± 10.2 a	11.6 ± 8.9 a	7.2 ± 9.4 a	33.8 ± 8.3 a	61.9 ± 4.5 a	34.7 ± 7.8 a

**Table 5 foods-14-01565-t005:** Fining effects of using different plant fibres in a continuous filtration system on the chromatic parameters and phenolic compounds of a red wine. Abbreviations: WGP1, white grape pomace fibre 1; WGP2, white grape pomace fibre 2; RGP, red grape pomace fibre; CoF, commercial fibre; CI, colour index; TP, total polyphenols; TA, total anthocyanins; PA, polymeric anthocyanins; TT, total tannins. Statistical analysis: ANOVA and LSD test (*p* < 0.05). Different letters indicate statistical differences.

	CI	TP	TA (mg/L)	PA (mg/L)	TT (mg/L)
**Control**	12.3 ± 0.1 cd	85.7 ± 0.1 d	220.3 ± 3.1 c	35.8 ± 0.2 c	2359.6 ± 14.2 c
WGP1F1	11.7 ± 0.0	80.8 ± 0.4	207.0 ± 5.7	34.0 ± 0.2	1829.5 ± 5.2
WGP1F2	12.3 ± 0.1	85.3 ± 0.6	217.5 ± 0.7	35.3 ± 0.5	1789.3 ± 4.8
WGP1F3	12.3 ± 0.0	84.4 ± 0.3	213.5 ± 7.8	35.5 ± 0.4	1873.4 ± 3.4
WGP1F4	12.3 ± 0.1	84.1 ± 0.1	221.5 ± 2.1	35.5 ± 0.4	1767.7 ± 36.9
**WGP1 mean value**	12.1 ± 0.0 b	83.6 ± 0.4 b	214.9 ± 4.1 b	35.1 ± 0.4 b	1815 ± 12.6 b
RGPF1	11.7 ± 0.0	80.3 ± 0.7	201.0 ± 1.4	33.9 ± 0.7	1744.8 ± 33.8
RGPF2	12.3 ± 0.1	83.8 ± 1.0	220.0 ± 0.0	35.2 ± 0.5	1803.3 ± 10.0
RGPF3	12.4 ± 0.1	86.0 ± 1.6	218.0 ± 2.8	35.6 ± 0.5	1893.5 ± 17.2
RGPF4	12.4 ± 0.1	85.5 ± 0.3	217.0 ± 1.4	35.7 ± 0.4	1960.4 ± 99.4
**RGP mean value**	12.2 ± 0.1 bc	84.2 ± 0.9 b	214.0 ± 1.4 b	35.1 ± 0.5 b	1850.5 ± 40.1 b
CoFF1	10.7 ± 0.1	76.3 ± 0.6	185.0 ± 2.8	30.1 ± 0.3	1520.7 ± 101.7
CoFF2	12.3 ± 0.1	84.2 ± 0.5	210.0 ± 2.8	35.3 ± 0.1	1848.7 ± 27.7
CoFF3	12.6 ± 0.1	84.8 ± 0.4	217.5 ± 2.1	36.3 ± 0.5	1790.3 ± 51.0
CoFF4	12.6 ± 0.0	86.1 ± 0.5	216.5 ± 4.9	36.5 ± 0.6	1851.3 ± 10.0
**CoF mean value**	12.0 ± 0.1 a	82.8 ± 0.5 a	207.3 ± 3.2 a	34.5 ± 0.4 a	1752.7 ± 47.6 a
WGP2F1	11.7 ± 0.0	80.7 ± 0.8	201.0 ± 4.2	33.0 ± 0.2	1914.3 ± 46.7
WGP2F2	12.5 ± 0.1	85.2 ± 1.3	214.5 ± 4.9	35.9 ± 0.3	1857.1 ± 17.5
WGP2F3	12.6 ± 0.0	86.4 ± 0.1	211.5 ± 4.9	35.9 ± 0.0	1810.9 ± 54.4
WGP2F4	12.7 ± 0.1	86.5 ± 1.3	221.0 ± 2.8	36.0 ± 0.0	1802.1 ± 9.2
**WGP2 mean value**	12.4 ± 0.0 d	84.7 ± 0.9 c	212.0 ± 4.2 b	35.2 ± 0.1 b	1846.1 ± 32.0 b

## Data Availability

The data will be made available upon request.

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
