# Peer review of "Grape Pomace Fibres as a Sustainable Fining Agent to Ensure Red Wine Safety: A First Approach in a Continuous System"

_foods, 2025, doi:10.3390/foods14091565_

Round 1
Reviewer 1 Report
Comments and Suggestions for Authors
The manuscript studied the use of grape pomace fibers to fine to reduce contaminants in red wine. Here are my concerns-
(1) more details about the grape pomace fibers are necessary, such as the preparation, characterazation of the fibers.
(2) provide a detailed schematic in Figure 1, such as the collection of each fraction. why diatomaceous earth was layered with fibers and whether it contributed to adsorption.
(3) please rationalize OTA and histamine concentrations, and pesticide selection. Are they valid levels present and prevalent in real vineyards.
(4) Statistical analysis in the study shoudl be specified, and statistics in the tables should be clarified.
(5) more studies and discussions about mechanisms are necessary.
(6) Will the fluid dynamics have impact on the fining? please discuss the comparasion between static and continuous systems.
(7) more discussion is necessary to explain the differences between grape fibers and commercials.
(8) Clarify the Conflict of Interest. The authors declared "Agrovin has no actual financial interest", but also acknowledged the funding from Agrovin.
Author Response
Dear Reviewer, you will find all our answers to your suggestions in the attached file.

Reviewer 2 Report
Comments and Suggestions for Authors
The authors of the manuscript "Grape pomace fibres as a sustainable fining agent to ensure red wine safety. A first approach in a continuous system" undertook interesting research related to improving the quality of wine by reducing the content of undesirable substances such as OTA, histamine and selected pesticides. Accelerating the wine clarification process and improving its biological quality is a significant progress in the production of wine on a commercial scale. The obtained results are valuable research material, but they have been poorly discussed. The authors did not take into account in the discussion an important issue related to the sorption capacity of absorbents in the dynamic filtration system, in this case diatomaceous earth and fibres. It is clearly visible in the first table that the subsequent samples received had higher OTA contents. The authors should determine the sorption capacity of the bed depending on the filtered wine. Therefore, the discussion of the results should be carried out again. Separately for each sample of 250 ml taken, which should then be repeated at least three times. In this way it will be possible to obtain the sorption efficiency in each of the 4 tests. The authors should conduct the experiments until the samples from the last three times are similar. In this way it would be possible to calculate how much the bed is able to work effectively. In this case the graphic part should also be improved.
Author Response

(The authors gave the same response as above.)

Reviewer 3 Report
Comments and Suggestions for Authors
In this article, the authors sought to enhance the value of grape marc, a by-product of the wine industry. In particular, it was used as a clarifying agent to improve the safety of contaminated young red wine. Through a continuous filtration system, the grape marc fibres significantly reduced the concentration of ochratoxin A, histamine and pesticides, preserving the colour of the wine and reducing the clarification time from days to hours.The introduction of the article is well structured and provides adequate context on the importance of sustainability in the wine industry.The description of the materials and methods is clear.The results, although clearly presented and adequately discussed, need to be reviewed, especially in the table presentation.Below are my thoughts on how to improve the article and render it publishable in the journal:
Line 2: 2 in superscript.
Line 18: OTA is the acronym for ochratoxin A. Put it in brackets on line 17. It can be reported later as an acronym.
Lines 22-24: marc fiber.Add a sentence about the potential industrial applications of marc fiber.
Line 31: authors' guidelines are not followed when citing references in the text. For example, "(García Lomillo et al., 2017)" should be reported as [1]. Apply this throughout the text of the manuscript.
Line 34: "OIV"?
Lines 40-43: Include a brief literature review of existing clarification techniques and their limitations.
Line 82: In line 77, insert (BA) after biogenic amines. It is then possible to report the acronym BA as in line 82.
Lines 100-104: Add details on the preparation of marc fibres and their chemical characterisation.
Line 101: details of white grape varieties
Line 102: specify the red grape variety
Table 2, 3, 4 and 5: I think it is sufficient to report the mean and the standard deviation. I do not consider it useful to report the values for each replicate.
Lines 366-372: It would be helpful to include suggestions for further research and improvements based on the results obtained.
References: Format bibliographic citations according to the Author's Guidelines. Also missing is the meaning of the superscript letters that should be added to the caption of each table.
Author Response

(The authors gave the same response as above.)

Reviewer 4 Report
Comments and Suggestions for Authors
This manuscript examined the effectiveness of three pomace fibres and a commercial fibre in improving the safety of a young red wine, previously contaminated with ochratoxin A, histamine, and various pesticides, using a continuous filtration system. All the pomace fibres were capable of reducing OTA concentration around 50%, and one of the tested fibres exhibited a strong ability of decreasing most of the pesticides presented in the wine, the results even being better than when this fibre was used in static conditions. However, the paper is not acceptable in this form and the authors should make some major changes. Therefore, the work needs some relevant changes.
Line 8: Fix the character of (2).
Line 82: What is BA? You should write the full name or indicate what it is.
Paragraph 2.1: Indicate the location of the grape samples and why two white and one red variety were included in the study.
Line 147: To facilitate the reading of the article, the methodology and protocol followed for the analysis must be clearly stated in full.
Table 2-3-4-5: The statistical analysis (ANOVA) should be reported to evaluate the data.
The objectives of the study, its innovative aspects, and the rationale for conducting the work should be clearly stated in the introduction and conclusion.
Conclusion: The real-world scale applications of this study, particularly its implications for the wine industry and consumer health, should also be highlighted.
Author Response

(The authors gave the same response as above.)

Round 2
Reviewer 2 Report
Comments and Suggestions for Authors
I don't make any comments
Reviewer 4 Report
Comments and Suggestions for Authors
In my opinion, the manuscript is ready for publication. The authors have adequately addressed all the comments